# REINFORCEMENT LEARNING FOR CONTROL WITH PROBABILISTIC STABILITY GUARANTEE

## ABSTRACT

Reinforcement learning is promising to control dynamical systems for which the traditional control methods are hardly applicable. However, in control theory, the stability of a closed-loop system can be hardly guaranteed using the policy/controller learned solely from samples. In this paper, we will combine Lyapunov's method in control theory and stochastic analysis to analyze the mean square stability of MDP in a model-free manner. Furthermore, the finite sample bounds on the probability of stability are derived as a function of the number M and length T of the sampled trajectories. And we show that there is a lower bound on T and the probability is much more demanding for M than T. Based on the theoretical results, a REINFORCE-like algorithm is proposed to learn the controller and the Lyapunov function simultaneously.

## 1 INTRODUCTION

Reinforcement learning (RL) has achieved superior performance on some complicated control tasks (Kumar et al., 2016; Xie et al., 2019; Hwangbo et al., 2019) for which the traditional control engineering methods can be hardly applicable (Åström and Wittenmark, 1973; Morari and Zafiriou, 1989; Slotine et al., 1991). The dynamical system to be controlled is often highly stochastic and nonlinear which is typically modeled by Markov decision process (MDP), i.e.,

$$s_{t+1} \sim P(s_{t+1}|s_t, a_t), \forall t \in \mathbb{Z}_+ \tag{1}$$

where $s \in \mathcal{S} \subset \mathbb{R}^n$ denotes the state, $a \in \mathcal{A} \subset \mathbb{R}^m$ denotes the action and $P(s_{t+1}|s_t, a_t)$ is the transition probability function. An optimal controller can be learned from samples through "trial and error" by memorizing what has been experienced (Kaelbling et al., 1996; Bertsekas, 2019). However, there is a major caveat that prevents the real-world application of learning methods for control engineering applications. Without using a mathematical model, the current sample-based RL methods cannot guarantee the stability of the closed-loop system, which is the most important property of any control system as in control theory.

The most useful and general approach for studying the stability of a dynamical system is Lyapunov's method Lyapunov (1892), which is dominant in control engineering Jiang and Jiang (2012); Lewis et al. (2012); Boukas and Liu (2000). In Lyapunov's method, a suitable "energy-like" Lyapunov function $L(s)$ is selected and its derivative along the system trajectories is ensured to be negative semi-definite, i.e., $L(s_{t+1}) - L(s_t) < 0$ for all time instants and states, so that the state goes in the direction of decreasing the value of Lyapunov function and eventually converges to the origin or a sub-level set of the Lyapunov function.

In the traditional control engineering methods, a mathematical model must be given, i.e., the transition probability function in (1) is known. Thus the stability can be analyzed without the need to assess all possible trajectories. However, in learning methods, as the dynamic model is unknown, the "energy decreasing" condition has to be verified by trying out *all* possible consecutive data pairs in the state space, i.e., to verify infinite inequalities $L(s_{t+1}) - L(s_t) < 0$. Obviously, the "infinity" requirement makes it impractical to directly exploit Lyapunov's method in a model-free framework.

In this paper, we show that the mean square stability of the system can be analyzed based on a finite number of samples without knowing the model of the system. The contributions of this paper are summarized as follows:

1. Instead of verifying an infinite number of inequalities over the state space, it is possible to analyze the stability through a sampling-based method where only one inequality is needed.

2. Instead of using infinite sample pairs $\{s_{t+1}, s_t\}$, a finite-sample stability theorem is proposed to provide a probabilistic stability guarantee for the system, and the probability is an increasing function of the number $M$ and length $T$ of sampled trajectories and converging to 1 as $M$ and $T$ grow.

3. As an independent interest, we also derive the policy gradient theorem for learning stabilizing policy with sample pairs and the corresponding algorithm. We further reveal that the classic REINFORCE algorithm (Williams, 1992) is a special case of the proposed algorithm for the stabilization problem.

We also conclude two takeaways for the paper:

- Samples of a finite number $M$ and length $T$ of trajectories can be used for stability analysis with a certain probability. The probability is monotonically converging to 1 when $M$ and $T$ grow.
- There is a lower bound on $T$ and the probability is much more demanding for $M$ than $T$.
- The REINFORCE like algorithm can learn the controller and Lyapunov function simultaneously.

The paper is organized as follows: In Section 2, related works are introduced. In Section 3, the definition of mean-square stability (MSS) and the problem statement is given. In Section 4, the sample-based MSS theorem is proposed. In Section 5, we propose the probabilistic stability guarantee when only a finite number of samples are accessible and the probabilistic bound in a relation to the number and length of sampled trajectories is derived. In Section 6, based on the stability theorems, the policy gradient is derived and a model-free RL algorithm (L-REINFORCE) is given. Finally, a simulated Cartpole stabilization task is considered to demonstrate the effectiveness of the proposed method. In Section 7, the vanilla version of L-REINFORCE is tested on a simulated Cartpole stabilization task to demonstrate the effectiveness; it is further incorporated with the maximum entropy framework to control the more high-dimensional and stochastic systems, including a legged robot, HalfCheetah, and the molecular synthetic biological gene regulatory networks (GRN) corrupted by the additive and multiplicative uniform noises.

## 2   RELATED WORKS

**Lyapunov's Method** As a basic tool in control theory, the construction/learning of the Lyapunov function is not trivial and many works are devoted to this problem (Noroozi et al., 2008; Prokhorov, 1994; Serpen, 2005; Prokhorov and Feldkamp, 1999). In Perkins and Barto (2002), the RL agent controls the switch between designed controllers using Lyapunov domain knowledge so that any policy is safe and reliable. Petridis and Petridis (2006) proposes a straightforward approach to construct the Lyapunov functions for nonlinear systems using neural networks. Richards et al. (2018) proposes a learning-based approach for constructing Lyapunov neural networks with the maximized region of attraction. However, these approaches require the model of the system dynamics explicitly. Stability analysis in a model-free manner has not been addressed. In Berkenkamp et al. (2017), local stability is analyzed by validating the "energy decreasing" condition on discretized points in the subset of state space with the help of a learned model, meaning that only a finite number of inequalities need to be checked. This approach is further extended by using a Noise Contrastive Prior Bayesian RNN in Gallieri et al. (2019). Nevertheless, the discretization technique may become infeasible as the dimension and space of interest increases, limiting its application to rather simple and low-dimensional systems.

**Reinforcement Learning** In model-free reinforcement learning (RL), stability is rarely addressed due to the formidable challenge of analyzing and designing the closed-loop system dynamics by solely using samples Buşoniu et al. (2018), and the associated stability theory in model-free RL remains as an open problem Buşoniu et al. (2018); Gorges (2017). Recently, Lyapunov analysis is used in model-free RL to solve control problems with safety constraints Chow et al. (2018; 2019). In Chow et al. (2018), a Lyapunov-based approach for solving constrained Markov decision processes is proposed with a novel way of constructing the Lyapunov function through linear programming. In Chow et al. (2019), the above results were further generalized to continuous control tasks. It should be noted that even though Lyapunov-based methods were adopted in these results, neither of them addressed the stability of the system. In Postoyan et al. (2017), an initial result is proposed for the stability analysis of deterministic nonlinear systems with optimal controller for infinite-horizon discounted cost, based on the assumption that discount is sufficiently close to 1. However, in practice, it is rather difficult to guarantee the optimality of the learned policy unless certain assumptions on

the system dynamics are made Murray et al. (2003); Abu-Khalaf and Lewis (2005); Jiang and Jiang (2015). Furthermore, the exploitation of multi-layer neural networks as function approximations Mnih et al. (2015); Lillicrap et al. (2015) only adds to the impracticality of this requirement.

Given certain information on the model, Adaptive dynamic programming (ADP) can guarantee convergence to the optimal solution, and thus stability is naturally ensured Balakrishnan et al. (2008). For nonlinear systems with input-affine structure, model-free ADP algorithms can guarantee the stability of the closed-loop system Murray et al. (2003); Abu-Khalaf and Lewis (2005); Shih et al. (2007); Jiang and Jiang (2015); Deptula et al. (2018). This paper steps beyond the scope of control-affine systems and are devoted to learning a controller with a stability guarantee for the general stochastic nonlinear system. To the best of the author's knowledge, the finite sample-based approach for the stability analysis of stochastic nonlinear systems considered in this paper is still missing.

For the model-based approaches, promising results on stability analysis are reported but generally based on certain model assumptions. Model predictive control (MPC) has long been studying the issue of optimal control of various dynamical systems without violating state and action constraints, and Lyapunov stability is naturally guaranteed (Mayne and Michalska, 1990; Michalska and Mayne, 1993; Mayne et al., 2000). Favorable as it may seem, the nice properties above are built upon the accurate and concise modeling of the dynamics, which narrows its scope to certain fields. In Ostafew et al. (2014), a learning-based nonlinear MPC algorithm is proposed to learn the disturbance model online and improve the tracking performance of field robots, but first, *a priori* model is required. Aswani et al. (2013) proposed a new learning-based MPC scheme that can provide deterministic guarantees on robustness while performance is improved by identifying a richer model. However, it is limited to the case that a linear model with known uncertainty bound is available. Other results concerning learning-based MPC are referred to Aswani et al. (2011); Bouffard et al. (2012); Di Cairano et al. (2013).

In Bobiti (2017); Bobiti and Lazar (2018), a sampling-based approach for stability analysis and domain of attraction estimation is proposed for deterministic nonlinear systems. The reliability of the estimation is addressed with a probabilistic bound on the number of samples, however, based on the assumption that all the samples are independently distributed. This infers that given multiple state trajectories, only the first-step data are applicable for the stability analysis, which is inefficient in a model-free framework and will be improved in this paper. Nevertheless, the aforementioned approach can be favorable in a model-based setup (Gallieri et al., 2019), given that 1-step predictions can be performed in parallel. It should also be noted that this paper is to address the stability analysis and control of stochastic systems, while the results above are focused on the deterministic nonlinear systems.

## 3 PROBLEM STATEMENT

Before establishing any stability theorem, the definition of stability needs to be properly given. In this paper, we will focus on the mean square stability (MSS) which is commonly known in control theory. The definition of MSS is given as follows.

**Definition 1** *(Shaikhet, 1997) The stochastic system is said to be mean square stable (MSS) if there exists a positive constant $b$ such that $\lim_{t\to\infty} \mathbb{E}_{s_t}\|s_t\|_2^2 = 0$ holds for any initial condition $s_0 \in \{s_0 | \|s_0\|_2^2 \le b\}$. If $b$ is arbitrarily large then the stochastic system is globally mean square stable (GMSS).*

MSS basically says that, on average, the state of a system, starting from an initial position in the state space, tends towards the equilibrium as time goes to infinity. It should be noted that the stability conditions of Markov chains have been reported in (Shaikhet, 1997; Meyn and Tweedie, 2012), however, of which the validation requires verifying infinite inequalities on the state space if $\mathcal{S}$ is continuous. Unfortunately, the finite sample-based approach for stability analysis where only one inequality needs to be checked is still missing.

For the sample-based approach, the key challenge is the theoretical gap in "finity" guarantees, i.e., (1) from infinite $L(s_{t+1}) - L(s_t)$ to only a single inequality related to sample expectation; (2) from infinite samples expectation to finite samples expectation. Thus in this paper, two sets of theoretical questions need to be answered.

**Q1.** What does a sample-based Lyapunov theorem look like and what are the assumptions and conditions needed to use a single $E_{\text{infinite samples}}(L(s_{t+1}) - L(s_t))$ instead of the infinite $L(s_{t+1}) - L(s_t)$?

**Q2.** What will be the number of samples needed to guarantee stability for a given probability, if $E_{\text{infinite samples}}$ is changed to $E_{\text{finite samples}}$? What is the analytical form of the probability as a function of the number $M$ and length $T$ of the sampled trajectories?

Before proceeding, some notations are to be defined. We introduce $c(s) \triangleq \min(\|s\|_2^2, \overline{c}), \overline{c} > 0$ to denote the clipped norm of state. The closed-loop transition probability is denoted as $P_\pi(s'|s) \triangleq \int_{\mathcal{A}} \pi(a|s)P(s'|s,a)\mathrm{d}a$. We also introduce the closed-loop state distribution at a certain instant $t$ as $P(s|\rho, \pi, t)$, which could be defined iteratively: $P(s'|\rho, \pi, t+1) = \int_{\mathcal{S}} P_\pi(s'|s)P(s|\rho, \pi, t)\mathrm{d}s, \forall t \in \mathbb{Z}_{[1,\infty)}$ and $P(s|\rho, \pi, 1) = \rho(s)$, where $\rho(s)$ is the starting state distribution.

## 4 SAMPLE-BASED LYAPUNOV STABILITY GUARANTEE

In this section, we will answer **Q1** in Section 3 and present the key results on sample-based stability analysis. We will show that only a single inequality $E_{\text{infinite samples}}(L(s_{t+1}) - L(s_t)) \leq 0$ is enough for the verification of stability. First, we make the following assumption which is commonly exploited by many RL literature (Sutton et al., 2009; Korda and La, 2015; Bhandari et al., 2018; Zou et al., 2019).

**Assumption 1** *The Markov chain induced by policy $\pi$ is ergodic.*

It follows that there exists a unique stationary distribution $q_\pi(s) = \lim_{t\to\infty} P(s|\rho, \pi, t)$. The verification of ergodicity is in general an open question in practice. There are many systems proved to be ergodic in physics, statistic mechanics, economics, e.g. gambling games Peters (2019), the Anosov flow Anosov (2010), and dynamical billiards Park (2014), etc. The study of ergodicity of various systems and its verification composed a major branch of mathematics. If the transition probability is known for all states, the validation is possible but requires a large source of computation power to enumerate through the state space. As a matter of fact, the existence of the stationary state distribution is generally assumed to hold in the RL literature Melo et al. (2008); Levin and Peres (2017); Bhandari et al. (2018); Zou et al. (2019). In this paper, we focus on analyzing the stability of such systems with a probabilistic bound, as well as developing an algorithm to find stabilizing controllers.

In Definition 1, stability is defined in relation to the set of starting states, which is also called the region of attraction (ROA). If the MSS system starts within the ROA, its trajectory will be surely attracted to the equilibrium. To build a sample-based stability guarantee, we need to ensure that the states in ROA are accessible for the stability analysis. Thus the following assumption is made to ensure that every state in ROA has a chance to be sampled.

**Assumption 2** *There exists a positive constant $b$ such that $\rho(s) > 0, \forall s \in \{s|c(s) \leq b\}$.*

Based on the above assumptions, we can exploit Lyapunov's method to prove the sample-based stability theorem. In Lyapunov's method, a positive definite function called Lyapunov function is needed. The selection of the Lyapunov function is not trivial and largely determines the result of stability analysis. In this paper, we construct the Lyapunov function using the following parameterization,

$$L(s) = (f_\phi(s) - f_\phi(0))^2 + \sigma c(s) \tag{2}$$

where $f_\phi(s)$ is a fully-connected neural network (NN) with ReLU activation function. $\phi$ denotes the parameters of the network and $\sigma$ is a small positive constant.

**Theorem 1** *The stochastic system (1) is mean square stable if there exists a function $L : \mathcal{S} \to \mathbb{R}_+$ and positive constants $\alpha_1$, $\alpha_2$ and $\alpha_3$, such that*

$$\alpha_1 c(s) \leq L(s) \leq \alpha_2 c(s) \tag{3}$$

$$\mathbb{E}_{s\sim\mu_\pi}(\mathbb{E}_{s'\sim P_\pi} L(s') - L(s) + \alpha_3 c(s)) \leq 0 \tag{4}$$

*where*

$$\mu_\pi(s) \triangleq \lim_{T\to\infty} \frac{1}{T} \sum_{t=1}^{T} P(s|\rho, \pi, t)$$

*is the infinite sampling distribution (ISD).*

**Proof:** The proof can be found in Section A in the Appendix. The general idea of the proof will be summarized in the following. First, we prove that ISD $\mu_\pi$ exists if $q_\pi$ exists. Then we exploit the Abelian theorem and Egorov theorem to prove that $L(s_t)$ converges to zero at the infinite instant. Finally, (3) establishes the relation between $L(s_t)$ and $c(s_t)$ and concludes the proof.

**Remark 1** *For the Lyapunov function in (2), the value of $\alpha_2$ can be approximately estimated. In practice, we are typically concerned with the stability in a finite space $\mathcal{S}$ where the $\|s\|_2^2 \leq \bar{c}$ and $c(s) = \|s\|_2^2$. Thus $\max_s |f_\phi(s) - f_\phi(0)|^2/\|s\|_2^2 + \sigma$ is a valid choice for $\alpha_2$. Considering that the neural network $f_\phi$ with Relu activation is Lipschitz continuous, it follows that $\alpha_2 = L_f + \sigma$ where the Lipschitz constant $L_f$ can be efficiently estimated by using approaches in the literature (Scaman and Virmaux, 2018; Fazlyab et al., 2019; Zou et al., 2020).*

It can be found that in Theorem 1, the infinite number of energy decreasing conditions are replaced by only a single sample-based inequality (4). However, the validation of stability through a sample-based approach comes with a cost: it theoretically requires a tremendous, if not infinite, number of samples to thoroughly estimate the state distributions at instants from 0 to infinity, which is impractical.

Theorem 1 is valid for both model-free and model-based approach since the sample-based energy decreasing condition is aimed at canceling the requirement of point-wise energy decreasing condition. Nevertheless, in the model-free setting, the estimation of transition probability in (4) only adds to the complexity of sampling. In the next section, we will show that a finite number of samples should be informative enough to guarantee stability with a certain probability. More specifically, the probabilistic stability bound will be given by closing the gap between infinite and finite-sample guarantees.

## 5    FINITE SAMPLE PROBABILISTIC STABILITY BOUND

In this section, we will answer **Q2** in Section 3 and present the finite sample-based stability theorem.

To estimate the $\mu_\pi$ in Theorem 1, an infinite number of trajectories of infinite time steps are needed, whereas in practice only $M$ trajectories of $T$ time steps are accessible. Thus in this section, we will first introduce the finite-time sampling distribution (FSD) $\mu_\pi^T \triangleq \frac{1}{T}\sum_{t=1}^{T} P(s|\rho, \pi, t)$, as an intermediate to study the effect of the finite sample-based estimation. Apparently, $\lim_{T\to\infty} \mu_\pi^T = \mu_\pi$.

The general idea of exploiting $\mu_\pi^T$ is: we first derive the deviation of $\mathbb{E}_{\mu_\pi^T}\Delta L(s)$ from $\mathbb{E}_{\mu_\pi}\Delta L(s)$ with respect to $T$, where

$$\Delta L(s) \triangleq \mathbb{E}_{s'\sim P_\pi}L(s') - L(s) + \alpha_3 c(s)$$

then we study the effect of estimating $\mathbb{E}_{\mu_\pi^T}\Delta L(s)$ with sample average and derive the probabilistic bound. Finally, the above effects are unified to propose the finite sample-based stability guarantee.

Now, we first close the first gap in terms of deviation between $\mathbb{E}_{\mu_\pi^T}\Delta L(s)$ and $\mathbb{E}_{\mu_\pi}\Delta L(s)$. To quantitatively analyze this effect, we introduce the following assumption,

**Assumption 3** *There exist a constant $\gamma \in (0,1)$ such that for any $\pi$*

$$\sum_{t=1}^{T} \|P(s|\pi,\rho,t) - q_\pi(s)\|_1 \leq 2T^\gamma, \forall\, T \in \mathbb{Z}_+ \tag{5}$$

*where $\|P(s|\pi,\rho,t) - q_\pi(s)\|_1$ denotes the $L_1$-distance between probability measures $P$ and $Q$.*

**Remark 2** *It should be noted that the assumption above is not strict and should be generally easy to satisfy for ergodic MDPs. Because $q_\pi$ denotes the stationary state distribution, it naturally follows that $\sum_{t=1}^{T} \|P(s|\pi,\rho,t) - q_\pi(s)\|_1 \leq 2T^{\gamma(T)} \leq 2T$, where $\gamma(T) \in [0,1]$ without any further assumption. The assumption proposed here merely replaces this time-varying $\gamma(T)$ with a constant. As a matter of fact, uniform ergodic for irreducible and aperiodic Markov chains (Levin and Peres, 2017; Bhandari et al., 2018; Zou et al., 2019) is a special case of the above assumption, where the state distribution is required to converge to $q_\pi$ exponentially at the rate of $\gamma^t$. Nevertheless, Assumption 3 allows us to give the quantitative bound for the deviation between $\mathbb{E}_{\mu_\pi}\Delta L(s)$ and $\mathbb{E}_{\mu_\pi^T}\Delta L(s)$ with respect to $T$.*

Based on Assumption 3, we introduce the following Lemma.

**Lemma 1** *Let $T$ denotes the length of trajectories (also known as episodes or sequences). If there exist positive constants $\alpha_1$, $\alpha_2$ such that (3) hold, then*

$$\left| \mathbb{E}_{\mu_\pi} \Delta L(s) - \mathbb{E}_{\mu_\pi^T} \Delta L(s) \right| \leq 2\bar{c}(\alpha_3 + \alpha_2)T^{\gamma-1} \tag{6}$$

**Proof:** The proof can be found in Section B in the Appendix. As shown in (6), the deviation of finite-time estimation of $\Delta L(s)$ from the infinite time estimation decreases as $T$ grows and converges to zero if $T$ is infinity

In the following, we will derive the probabilistic bound on estimating $\mathbb{E}_{\mu_\pi^T} \Delta L(s)$ with $M$ trajectories of $T$ steps. It is worth mentioning that since $M$ trajectories are independent from each other, each trajectory as a whole is applicable for the estimation of $\Delta L(s)$ under $\mu_\pi^T$. This will be demonstrated in the following lemma, where increasing $M$ is desirable for the reduction of estimation deviation, while $T$ doesn't effect the probabilistic bound.

**Lemma 2** *Let $M$ denote the number of trajectories and $T$ denote the length of trajectories. If there exist positive constants $\alpha_1$, $\alpha_2$ such that (3) hold, then $\forall \beta \geq 0$,*

$$\mathbb{P}\left( \frac{1}{MT} \sum_{t=1}^{T} \sum_{m=1}^{M} \left( L(s_{t+1,m}) - L(s_{t,m}) + \alpha_3 c(s_{t,m}) \right) - \mathbb{E}_{\mu_\pi^T} \Delta L(s) \leq -\beta \right) \leq \exp\left( -\frac{2M\beta^2}{(2\alpha_2 + \alpha_3)^2 \bar{c}^2} \right) \tag{7}$$

*where $s_{t,m}$ denotes the sampled state in the $m$-th trajectory at time $t$.*

**Proof:** The proof can be found in Section C in the Appendix. A noteworthy fact is that $L(\cdot)$ in (7) is bounded since (3) hold and $c$ is a positive semi-definite variable clipped by $\bar{c}$. Thus it is straightforward to apply Hoeffding's inequality to derive the probabilistic bound.

Now, the finite sample estimation of $\Delta L(s)$ and $\mathbb{E}_{\mu_\pi} \Delta L(s)$ are connected with $\mathbb{E}_{\mu_\pi^T} \Delta L(s)$ respectively by Lemma 1 and 2, we will unify them to derive the desired probabilistic stability guarantee.

**Theorem 2** *If there exists a function $L : \mathcal{S} \to \mathbb{R}_+$ and positive constants $\alpha_1$, $\alpha_2$ and $\alpha_3$, such that (3) hold, and for a number of $M$ trajectories with $T$ time steps there exists a positive constant $\epsilon$ such that*

$$T \geq \left( \frac{b_1}{\epsilon} \right)^{\frac{1}{1-\gamma}} \tag{8}$$

$$\frac{1}{MT} \sum_{t=1}^{T} \sum_{m=1}^{M} \left( L(s_{t+1,m}) - L(s_{t,m}) + \alpha_3 c(s_{t,m}) \right) \leq -\epsilon \tag{9}$$

*then the stochastic system can be guaranteed to be mean square stable with probability at least*

$$\mathbb{P}\left( \mathbb{E}_{\mu_\pi} \Delta L(s) \leq 0 \right) \geq 1 - \exp\left( -2M \left( \frac{\epsilon - T^{\gamma-1} b_1}{b_2} \right)^2 \right) \tag{10}$$

*where $b_1 = 2(\alpha_3 + \alpha_2)\bar{c}$ and $b_2 = (2\alpha_2 + \alpha_3)\bar{c}$. If the desired confidence of stability guarantee is at least $\delta$, the associated overall sample complexity is at least $\mathcal{O}(\log(\frac{1}{1-\delta}))$. To achieve a confidence $\delta$, $M$ and $T$ have to satisfy $M(\epsilon - T^{\gamma-1} b_1)^2 \geq 2\bar{c}^2 (2\alpha_2 + \alpha_3)^2 \log(\frac{1}{1-\delta})$.*

**Proof:** The proof can be found in Section D in the Appendix. The idea is that we estimate $\Delta L(s)$ with finite samples in (9) and strengthen this finite sample-based condition with a constant $\epsilon$, such that a small deviation in estimation will not cause misjudgment of stability. In practice, $\epsilon$ is a hyperparameter to be tuned according to the number of samples available.

**Remark 3** *In (Kearns and Singh, 2002; Strehl et al., 2006; Jin et al., 2018), the finite-sample analysis and asymptotic convergence of various classical RL algorithms have been extensively studied. However, to the best of our knowledge, Theorem 2 is the first finite-sample analysis for sample-based stability analysis, providing a probabilistic stability guarantee that is related to the number of samples. The probabilistic bound (10) is a monotone increasing function of $T$ and $M$,*

*and approaches to 1 as $T$ and $M$ tend to infinity. Intuitively, the trajectories with inadequate length cannot reflect the evolution of the system dynamics and thus are not applicable in the stability analysis. Thus the requirement that the length of the trajectories $T$ be greater than a minimum value (8) is reasonable. Nevertheless, it is possible to derive tighter bounds in the future by applying other inequalities, such as Bernstein inequality. The sharpness of the derived bound will be illustrated combined with a Cartpole example in Section 7.*

## 6   SAMPLE-BASED CONTROL WITH STABILITY GUARANTEE

Based on the theoretical results in the previous sections, one can judge whether the system is stable given several finite-length trajectories by estimating (9). The theoretical results in the stability theorems using Lyapunov's method do not, however, give a prescription for determining the Lyapunov function and controller. To translate the theorem into practical algorithms, the high-level plan is to parameterize $L(s)$ with (2) and the controller $\pi(a|s)$ with an arbitrary NN $\pi_\theta(a|s)$. Then $\phi$ and $\theta$ will be updated separately and iteratively using stochastic gradient descent algorithms until system (1) is stabilized such that (9) is satisfied. We use $\tau$ to denote a trajectory ($\tau = \{s_1, a_1, s_2...s_T\}$), and $\tau \sim \pi$ is the shorthand for indicating that the distribution over trajectories depends on $\pi$: $P(\tau) = \rho(s_1) \prod_{t=1}^{T} P(s_{t+1}|s_t, a_t)\pi(a_t|s_t)$.

### 6.1   POLICY GRADIENT

In this subsection, we will focus on how to learn the controller in an iterative manner, repeatedly estimating the policy gradient of the target function with samples and updating $\theta$ through stochastic gradient descent. $\Delta L(s)$ is temporarily assumed to be given, i.e., $\phi$ are fixed. In Section 6.2, we will show how the Lyapunov function is selected and learned after $\theta$ is updated.

Since the left-hand side of (9) is the unbiased estimate of $\Delta L(s)$ on $\mu_\pi^T$, the problem can be formulated by

$$\text{find } \theta, \text{ s.t. } \mathbb{E}_{\mu_{\pi_\theta}^T} \Delta L(s) \leq -\epsilon \tag{11}$$

A straightforward way of solving the constrained optimization problem above would be the first-order method Bertsekas (2014) (Chapter 4), also known as gradient descent. At each update step, the gradient of (11) with respect to $\theta$ is estimated with samples, and $\theta$ updates a small step in the opposite direction of the estimated gradient vector. The gradient of (11) with respect to $\theta$ is derived in the following theorem.

**Theorem 3** *The gradient of Lyapunov condition (11) is given by the following,*

$$\nabla_\theta \mathbb{E}_{\mu_{\pi_\theta}^T} \Delta L(s) = \mathbb{E}_{\tau \sim \pi_\theta} \left[ \frac{1}{T} \sum_{t=1}^{T} \nabla_\theta \log \pi_\theta(a_t|s_t) \left[ \alpha_3 \mathcal{C}_{t+1:T} + L(s_{T+1}) \right] \right] \tag{12}$$

*where $\mathcal{C}_{t_1:t_2} = \sum_{t=t_1}^{t_2} c(s_t)$ denotes the sum of cost $c$ over a time interval and $\mathcal{C}_{t+1:t} = 0$*

The proof of Theorem 3 can be found in Section E in the Appendix.

Surprisingly, we found that the policy gradient derived in Theorem 3 is very similar to that used in the vanilla policy gradient method, i.e., REINFORCE Sutton and Barto (2018), in the classic RL paradigm. In RL, the objective is to minimize a certain objective function $J_\theta = \mathbb{E}_{\tau \sim \pi_\theta} \left[ \sum_{t=1}^{T} c(s_t) \right]$ and the policy gradient is given as follows:

$$\nabla_\theta J_\theta = \mathbb{E}_{\tau \sim \pi_\theta} \left[ \sum_{t=1}^{T} \nabla_\theta \log \pi_\theta(a_t|s_t) \mathcal{C}_{t+1:T+1} \right] \tag{13}$$

Essentially, despite the scale of $\frac{1}{T}$, (12) and (13) are equivalent if one chooses $c(s)$ to be the Lyapunov function and sets $\alpha_3 = 1$. This implies that given system (1), REINFORCE actually updates the policy towards a solution that can stabilize the system, although it is now aware of under what conditions the solution is guaranteed to be stabilizing. In particular, we can view REINFORCE as

a special case of our result, since we prove that many other choices of $\alpha_3$ and Lyapunov functions are admissible to find a stabilizing solution. The default setting of $c(s)$ as $L(s)$ and $\alpha_3 = 1$ in REINFORCE may not satisfy (9), while we reveled that many other feasible combinations of $L$ and $\alpha_3$ potentially exist.

In light of this connection with REINFORCE, it is natural to propose a similar learning procedure based on Theorem 3, which we name as Lyapunov-REINFORCE (termed as 'L-REINFORCE'). L-REINFORCE updates the policy with the policy gradient proposed in (12). Instead of minimizing the objective function (6.1), L-REINFORCE aims to learn a stochastic policy $\pi(a|s)$ such that the conditions in Theorem 2 are satisfied.

Furthermore, to reduce the variance in the estimation of (12) and speed up learning, it is desirable to introduce a baseline function $b(s)$ in (12) and the estimation is still unbiased Sutton and Barto (2018):

$$\nabla_\theta \mathbb{E}_{\mu_{\pi_\theta}^T} \Delta L(s) = \mathop{\mathbb{E}}_{\tau \sim \pi_\theta} \left[ \frac{1}{T} \sum_{t=1}^{T} \nabla_\theta \log \pi_\theta(a_t|s_t) \big[ \alpha_3 \mathcal{C}_{t+1:T} + L(s_{T+1}) - b(s_t) \big] \right]$$

### 6.2 LYAPUNOV FUNCTION

The Lyapunov function is parameterized using a DNN $f_\phi$ in (2). In fact, any real function $f$ is admissible in (2) to construct Lyapunov function, thus many ways of updating $\phi$ are applicable in our framework, e.g. Prokhorov (1994); Petridis and Petridis (2006); Richards et al. (2018). In this paper, we intuitively choose the value function to be the update target for $f$ to examine the effectiveness of proposed results as exploited in Berkenkamp et al. (2017); Chow et al. (2019) and leave other possible choices for future work.

To wrap up, the L-REINFORCE algorithm is summarized in Algorithm 1.

---

**Algorithm 1** L-REINFORCE

---

  **repeat**
    **for** 1, 2, ..., M **do**
      Collect transition pairs following $\pi_\theta$ for $T$ steps
    **end for**
    $\theta \leftarrow \theta - \alpha \nabla_\theta \mathbb{E}_{\mu_{\pi_\theta}^T} \Delta L(s)$
    Update $\phi$ of Lyapunov function/value network to approximate the designed target
  **until** There exists $\alpha_3$ such that $\mathbb{E}_{\mu_{\pi_\theta}^T} \Delta L(s) < -\epsilon$

---

## 7 EXPERIMENT

In this section, two sets of experiments are conducted. First, the vanilla L-REINFORCE algorithm is evaluated on a Cartpole example with comparison to REINFORCE and soft actor-critic (SAC). Then, to further demonstrate the effectiveness of the proposed framework on more complicated systems, L-REINFORCE is incorporated with the maximum entropy method and tested in the more high-dimensional and stochastic systems.

### 7.1 A CARTPOLE EAXAMPLE

To demonstrate the effectiveness of the proposed method, we consider the stabilization task of a simulated Cartpole Brockman et al. (2016). The goal is to stabilize the pole vertically at the position $x = 0$. We adopt REINFORCE as the baseline method for comparison. In addition, soft actor-critic (SAC) Haarnoja et al. (2018), a state-of-the-art off-policy RL algorithm, is also included. L-REINFORCE and REINFORCE select the action among $\{-10, 0, 10\}$. SAC selects the control force in the continuous space with the same minimal and maximal value, thus better performance can be potentially achieved. The detailed experiment setup and hyperparameters are presented in Appendix F.

It is important to note that the stability of a system can not be judged from the cumulative cost (or return) because stability is an inherent property of the system dynamics and a stable system may not be optimal in terms of the return. Thus in Figure 1, we show the transient system behavior under the

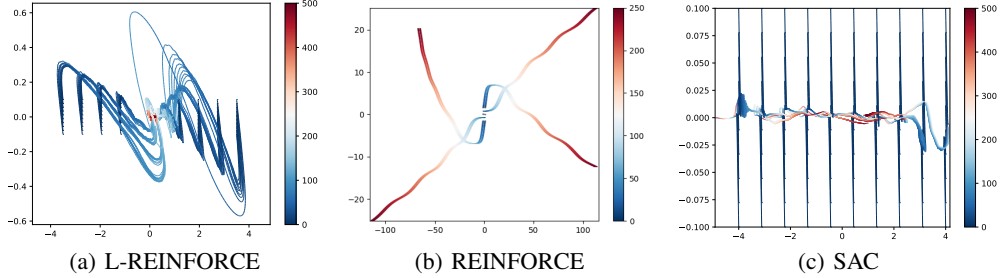

| (a) L-REINFORCE | (b) REINFORCE | (c) SAC |
|---|---|---|

Figure 1: Phase trajectories of the agents trained by L-REINFORCE, REINFORCE, and SAC. The X-axis denotes the position $x$ and the Y-axis denotes the angle $\theta$ in rads. The trajectories are of 500 timesteps and the states at different instants are indicated by respective colors, corresponding to the color-bar to the right.

learned policies of L-REINFORCE, REINFORCE, and SAC. The agents are initialized at different positions in the space and their subsequent behaviors are observed. As shown in Figure 1, starting from different initial states, L-REINFORCE can efficiently stabilize the system to the origin. On the contrary, the cartpole diverges under the control of REINFORCE (Figure 1. b). SAC can also keep the pole vertically, but the cart can not be stabilized to the position where $x = 0$. In some cases, the SAC agent even slowly drifts away (see the left side of Figure 1. c). To have a more clear view, these trajectories are further illustrated in the time domain in Appendix F.

To let the readers have an intuitive sense of the probabilistic stability bound, the bound for Cartpole under the control of the L-REINFORCE agent is shown in Figure 2. As shown in Figure 2, the probability of stability increases sharply as the minimum $T$ requirement (8) is satisfied. Increasing $M$ and $T$ are both helpful for raising the confidence of stability guarantee. The probabilistic stability guarantee measures the reliability of learned policies. By tuning the hyperparameters such as $M, T$, and $\alpha_3$, one can achieve the confidence of stability guarantee according to the real needs.

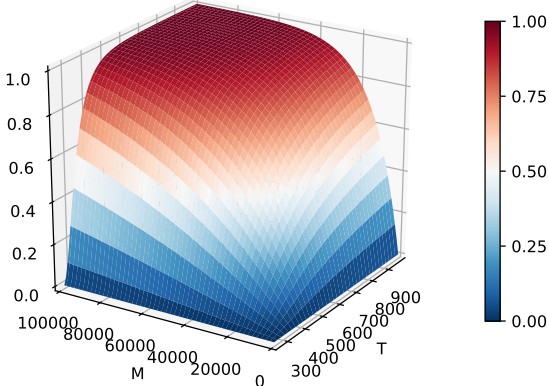

Figure 2: Visualization of the probabilistic stability bound. The X-axis indicates the length of trajectories $T$ and Y-axis indicates the number of episodes $M$. The Z-axis indicates the probability of stability and the values are colored differently according to the color-bar.

## 7.2 HIGH DIMENSIONAL EXAMPLES

In this part, we will illustrate the effectiveness of the proposed framework on some high dimensional control problems, where the system dynamics are highly nonlinear and even corrupted by various noises, thus making them more stochastic and challenging. Three examples are included: a high-dimensional continuous control problem of 3D robots, HalfCheetah, the molecular synthetic biological gene regulatory networks (GRN) corrupted by the additive and multiplicative uniform noises. In the living cells of biological systems, gene expression is very noisy and there are strong evidence on the genetic basis on these noises in literature of genetic regulatory networks Swain et al. (2002); Bar-Even et al. (2006). Details of the experimental setup are referred to the Appendix.

To achieve high performance on these continuous control tasks, we further incorporate the maximum entropy method (Shi et al., 2019; Haarnoja et al., 2018; Zhao et al., 2019) in the proposed framework. By introducing the entropy regularization, the policy is encouraged to explore more and less easy to early convergence to suboptimal solutions. To have a fair comparison, only SAC is included as the baseline in these examples, given to its superior performance on continuous control tasksHaarnoja et al. (2018), while REINFORCE is excluded due to its poor performance. Implementation details of the algorithm is referred to the Appendix.

In Figure 3, 4 and 5, the state trajectories of the systems are shown in time-domain. It is observed that even though the systems are highly nonlinear and stochastic due to the noises, L-REINFORCE is still able to stabilize the tracking error to zero in the mean. In comparison, although SAC succeeded in stabilization in some of the trials, see Figure 4 and 5, but its success appears to be very random and can be hardly guaranteed.

## 8 CONCLUSION

In this paper, we proposed a sampling-based approach for stability analysis of nonlinear stochastic systems modeled by the Markov decision process in a model-free manner. Instead of verifying energy decreasing point-wisely on the state space, we proposed a stability theorem where only one sampling-based inequality is to be checked. Furthermore, we showed that with a finite number of trajectories of finite length, it is possible to guarantee stability with a certain probability and the probabilistic bound is derived. Finally, we proposed a model-free learning algorithm to learn the controller with a stability guarantee and revealed its connection to REINFORCE. REINFORCE is not the state-of-the-art RL algorithm for complicated continuous tasks. In the future, an important direction is to extend the theoretical analysis to more efficient algorithms than REINFORCE.

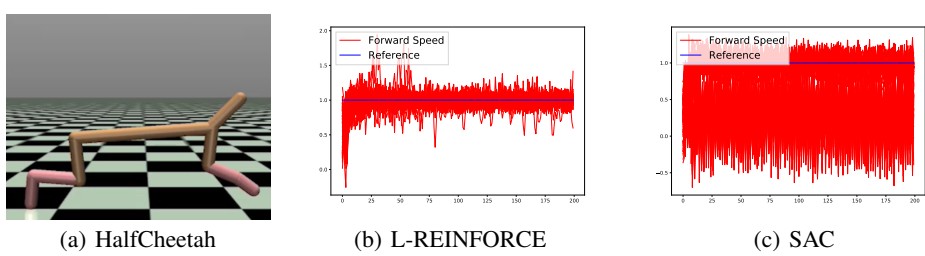

| (a) HalfCheetah | (b) L-REINFORCE | (c) SAC |

Figure 3: State trajectories of the agents trained by L-REINFORCE (b) and SAC (c). The X-axis denotes the time $t$ and the Y-axis denotes the forward speed of the robot. The task is to control the robot to run forward at the reference speed $1m/s$.

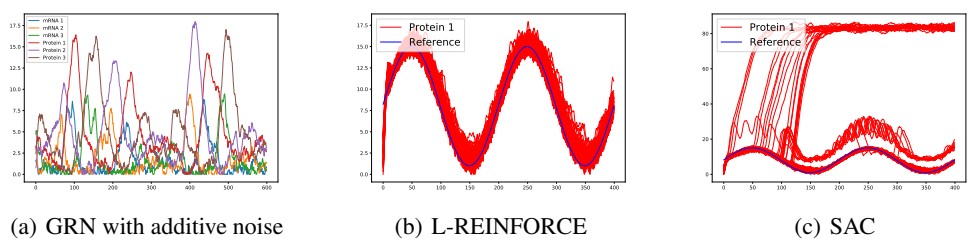

| (a) GRN with additive noise | (b) L-REINFORCE | (c) SAC |

Figure 4: State trajectories of the agents trained by L-REINFORCE (b) and SAC (c). (a) shows the uncontrolled dynamic of the GRN with additive uniform noises. The X-axis denotes the time $t$ and the Y-axis denotes the concentration of each component. The task is to control the concentration of Protein 1 to track a reference signal, which is a sine signal.

## REFERENCES

Vikash Kumar, Abhishek Gupta, Emanuel Todorov, and Sergey Levine. Learning dexterous manipulation policies from experience and imitation. *arXiv preprint arXiv:1611.05095*, 2016.

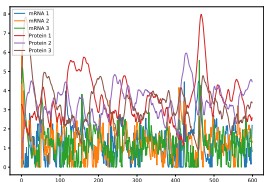 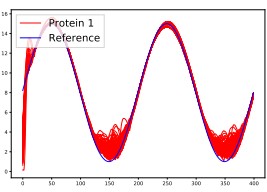 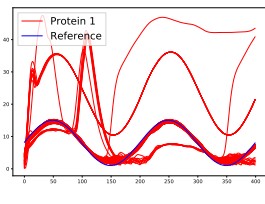

(a) GRN with multiplicative noise     (b) L-REINFORCE     (c) SAC

Figure 5: State trajectories of the agents trained by L-REINFORCE (b) and SAC (c). (a) shows the uncontrolled dynamic of the GRN with multiplicative uniform noises. The X-axis denotes the time $t$ and the Y-axis denotes the concentration of each component. The task is to control the concentration of Protein 1 to track a reference signal, which is a sine signal.

Zhaoming Xie, Patrick Clary, Jeremy Dao, Pedro Morais, Jonathan Hurst, and Michiel van de Panne. Iterative reinforcement learning based design of dynamic locomotion skills for cassie. *arXiv preprint arXiv:1903.09537*, 2019.

Jemin Hwangbo, Joonho Lee, Alexey Dosovitskiy, Dario Bellicoso, Vassilios Tsounis, Vladlen Koltun, and Marco Hutter. Learning agile and dynamic motor skills for legged robots. *Science Robotics*, 4(26):eaau5872, 2019.

Karl Johan Åström and Björn Wittenmark. On self tuning regulators. *Automatica*, 9(2):185–199, 1973.

Manfred Morari and Evanghelos Zafiriou. *Robust process control*. Morari, 1989.

Jean-Jacques E Slotine, Weiping Li, et al. *Applied nonlinear control*, volume 199. Prentice hall Englewood Cliffs, NJ, 1991.

Leslie Pack Kaelbling, Michael L Littman, and Andrew W Moore. Reinforcement learning: A survey. *Journal of Artificial Intelligence Research*, 4:237–285, 1996.

Dimitri Bertsekas. *Reinforcement Learning and Optimal Control*. Athena Scientific, 2019.

Aleksandr Mikhailovich Lyapunov. *The general problem of the stability of motion (in Russian)*. PhD Dissertation, Univ. Kharkov, 1892.

Yu Jiang and Zhong-Ping Jiang. Computational adaptive optimal control for continuous-time linear systems with completely unknown dynamics. *Automatica*, 48(10):2699–2704, 2012.

Frank L Lewis, Draguna Vrabie, and Vassilis L Syrmos. *Optimal control*. John Wiley & Sons, 2012.

EK Boukas and ZK Liu. Robust stability and $H_\infty$ control of discrete-time jump linear systems with time-delay: an LMI approach. In *Decision and Control, 2000. Proceedings of the 39th IEEE Conference on*, volume 2, pages 1527–1532. IEEE, 2000.

Ronald J Williams. Simple statistical gradient-following algorithms for connectionist reinforcement learning. *Machine learning*, 8(3-4):229–256, 1992.

Navid Noroozi, Paknoosh Karimaghaee, Fatemeh Safaei, and Hamed Javadi. Generation of Lyapunov functions by neural networks. In *Proceedings of the World Congress on Engineering*, volume 2008, 2008.

Danil V Prokhorov. A Lyapunov machine for stability analysis of nonlinear systems. In *Proceedings of 1994 IEEE International Conference on Neural Networks (ICNN'94)*, volume 2, pages 1028–1031. IEEE, 1994.

Gursel Serpen. Empirical approximation for Lyapunov functions with artificial neural nets. In *Proceedings. 2005 IEEE International Joint Conference on Neural Networks, 2005.*, volume 2, pages 735–740. IEEE, 2005.

Danil V Prokhorov and Lee A Feldkamp. Application of svm to Lyapunov function approximation. In *IJCNN'99. International Joint Conference on Neural Networks. Proceedings (Cat. No. 99CH36339)*, volume 1, pages 383–387. IEEE, 1999.

Theodore J Perkins and Andrew G Barto. Lyapunov design for safe reinforcement learning. *Journal of Machine Learning Research*, 3(Dec):803–832, 2002.

Vassilios Petridis and Stavros Petridis. Construction of neural network based lyapunov functions. In *The 2006 IEEE International Joint Conference on Neural Network Proceedings*, pages 5059–5065. IEEE, 2006.

Spencer M Richards, Felix Berkenkamp, and Andreas Krause. The Lyapunov neural network: Adaptive stability certification for safe learning of dynamical systems. In *Conference on Robot Learning*, pages 466–476, 2018.

Felix Berkenkamp, Matteo Turchetta, Angela Schoellig, and Andreas Krause. Safe model-based reinforcement learning with stability guarantees. In *Advances in neural information processing systems*, pages 908–918, 2017.

Marco Gallieri, Seyed Sina Mirrazavi Salehian, Nihat Engin Toklu, Alessio Quaglino, Jonathan Masci, Jan Koutník, and Faustino Gomez. Safe interactive model-based learning. *arXiv preprint arXiv:1911.06556*, 2019.

Lucian Buşoniu, Tim de Bruin, Domagoj Tolić, Jens Kober, and Ivana Palunko. Reinforcement learning for control: Performance, stability, and deep approximators. *Annual Reviews in Control*, 2018.

Daniel Gorges. Relations between model predictive control and reinforcement learning. *IFAC-PapersOnLine*, 50(1):4920–4928, 2017.

Yinlam Chow, Ofir Nachum, Edgar Duenez-Guzman, and Mohammad Ghavamzadeh. A Lyapunov-based approach to safe reinforcement learning. *arXiv preprint arXiv:1805.07708*, 2018.

Yinlam Chow, Ofir Nachum, Aleksandra Faust, Mohammad Ghavamzadeh, and Edgar Duenez-Guzman. Lyapunov-based safe policy optimization for continuous control. *arXiv preprint arXiv:1901.10031*, 2019.

Romain Postoyan, Lucian Buşoniu, Dragan Nešić, and Jamal Daafouz. Stability analysis of discrete-time infinite-horizon optimal control with discounted cost. *IEEE Transactions on Automatic Control*, 62(6):2736–2749, 2017.

John J Murray, Chadwick J Cox, and Richard E Saeks. The adaptive dynamic programming theorem. In *Stability and control of dynamical systems with applications*, pages 379–394. Springer, 2003.

Murad Abu-Khalaf and Frank L Lewis. Nearly optimal control laws for nonlinear systems with saturating actuators using a neural network hjb approach. *Automatica*, 41(5):779–791, 2005.

Yu Jiang and Zhong-Ping Jiang. Global adaptive dynamic programming for continuous-time nonlinear systems. *IEEE Transactions on Automatic Control*, 60(11):2917–2929, 2015.

Volodymyr Mnih, Koray Kavukcuoglu, David Silver, Andrei A Rusu, Joel Veness, Marc G Bellemare, Alex Graves, Martin Riedmiller, Andreas K Fidjeland, Georg Ostrovski, et al. Human-level control through deep reinforcement learning. *Nature*, 518(7540):529–533, 2015.

Timothy P Lillicrap, Jonathan J Hunt, Alexander Pritzel, Nicolas Heess, Tom Erez, Yuval Tassa, David Silver, and Daan Wierstra. Continuous control with deep reinforcement learning. *arXiv preprint arXiv:1509.02971*, 2015.

SN Balakrishnan, Jie Ding, and Frank L Lewis. Issues on stability of adp feedback controllers for dynamical systems. *IEEE Transactions on Systems, Man, and Cybernetics, Part B (Cybernetics)*, 38(4):913–917, 2008.

Peter Shih, B Kaul, Sarangapani Jagannathan, and J Drallmeier. Near optimal output-feedback control of nonlinear discrete-time systems in nonstrict feedback form with application to engines. In *2007 International Joint Conference on Neural Networks*, pages 396–401. IEEE, 2007.

Patryk Deptula, Joel A Rosenfeld, Rushikesh Kamalapurkar, and Warren E Dixon. Approximate dynamic programming: Combining regional and local state following approximations. *IEEE transactions on neural networks and learning systems*, 29(6):2154–2166, 2018.

David Q Mayne and Hannah Michalska. Receding horizon control of nonlinear systems. *IEEE Transactions on automatic control*, 35(7):814–824, 1990.

Hanna Michalska and David Q Mayne. Robust receding horizon control of constrained nonlinear systems. *IEEE Transactions on automatic control*, 38(11):1623–1633, 1993.

David Q Mayne, James B Rawlings, Christopher V Rao, and Pierre OM Scokaert. Constrained model predictive control: Stability and optimality. *Automatica*, 36(6):789–814, 2000.

Chris J Ostafew, Angela P Schoellig, and Timothy D Barfoot. Learning-based nonlinear model predictive control to improve vision-based mobile robot path-tracking in challenging outdoor environments. In *2014 IEEE International Conference on Robotics and Automation (ICRA)*, pages 4029–4036. IEEE, 2014.

Anil Aswani, Humberto Gonzalez, S Shankar Sastry, and Claire Tomlin. Provably safe and robust learning-based model predictive control. *Automatica*, 49(5):1216–1226, 2013.

Anil Aswani, Neal Master, Jay Taneja, David Culler, and Claire Tomlin. Reducing transient and steady state electricity consumption in hvac using learning-based model-predictive control. *Proceedings of the IEEE*, 100(1):240–253, 2011.

Patrick Bouffard, Anil Aswani, and Claire Tomlin. Learning-based model predictive control on a quadrotor: Onboard implementation and experimental results. In *2012 IEEE International Conference on Robotics and Automation*, pages 279–284. IEEE, 2012.

Stefano Di Cairano, Daniele Bernardini, Alberto Bemporad, and Ilya V Kolmanovsky. Stochastic mpc with learning for driver-predictive vehicle control and its application to hev energy management. *IEEE Transactions on Control Systems Technology*, 22(3):1018–1031, 2013.

Ruxandra Valentina Bobiti. *Sampling–driven stability domains computation and predictive control of constrained nonlinear systems*. PhD thesis, PhD thesis, 2017. URL https://pure. tue. nl/ws/files/78458403 . . . , 2017.

Ruxandra Bobiti and Mircea Lazar. Automated-sampling-based stability verification and doa estimation for nonlinear systems. *IEEE Transactions on Automatic Control*, 63(11):3659–3674, 2018.

L Shaikhet. Necessary and sufficient conditions of asymptotic mean square stability for stochastic linear difference equations. *Applied Mathematics Letters*, 10(3):111–115, 1997.

Sean P Meyn and Richard L Tweedie. *Markov chains and stochastic stability*. Springer Science & Business Media, 2012.

Richard S Sutton, Hamid R Maei, and Csaba Szepesvári. A convergent $o(n)$ temporal-difference algorithm for off-policy learning with linear function approximation. In *Advances in neural information processing systems*, pages 1609–1616, 2009.

Nathaniel Korda and Prashanth La. On td (0) with function approximation: Concentration bounds and a centered variant with exponential convergence. In *International Conference on Machine Learning*, pages 626–634, 2015.

Jalaj Bhandari, Daniel Russo, and Raghav Singal. A finite time analysis of temporal difference learning with linear function approximation. *arXiv preprint arXiv:1806.02450*, 2018.

Shaofeng Zou, Tengyu Xu, and Yingbin Liang. Finite-sample analysis for sarsa with linear function approximation. In *Advances in Neural Information Processing Systems*, pages 8665–8675, 2019.

Ole Peters. The ergodicity problem in economics. *Nature Physics*, 15(12):1216–1221, 2019.

D. V Anosov. Ergodic properties of geodesic flows on closed riemannian manifolds of negative curvature. In *Dokl Akad Nauk Sssr*, 2010.

Sun Woo Park. An introduction to dynamical billiards, 2014.

Francisco S Melo, Sean P Meyn, and M Isabel Ribeiro. An analysis of reinforcement learning with function approximation. In *Proceedings of the 25th international conference on Machine learning*, pages 664–671, 2008.

David A Levin and Yuval Peres. *Markov chains and mixing times*, volume 107. American Mathematical Soc., 2017.

Kevin Scaman and Aladin Virmaux. Lipschitz regularity of deep neural networks: analysis and efficient estimation, 2018.

Mahyar Fazlyab, Alexander Robey, Hamed Hassani, Manfred Morari, and George J. Pappas. Efficient and accurate estimation of lipschitz constants for deep neural networks, 2019.

Dongmian Zou, Radu Balan, and Maneesh Singh. On lipschitz bounds of general convolutional neural networks. *IEEE Transactions on Information Theory*, 66(3):1738–1759, Mar 2020. ISSN 1557-9654. doi: 10.1109/tit.2019.2961812. URL http://dx.doi.org/10.1109/TIT.2019.2961812.

Michael Kearns and Satinder Singh. Near-optimal reinforcement learning in polynomial time. *Machine learning*, 49(2-3):209–232, 2002.

Alexander L Strehl, Lihong Li, Eric Wiewiora, John Langford, and Michael L Littman. Pac model-free reinforcement learning. In *Proceedings of the 23rd international conference on Machine learning*, pages 881–888, 2006.

Chi Jin, Zeyuan Allen-Zhu, Sebastien Bubeck, and Michael I Jordan. Is q-learning provably efficient? In *Advances in Neural Information Processing Systems*, pages 4863–4873, 2018.

Dimitri P Bertsekas. *Constrained optimization and Lagrange multiplier methods*. Academic press, 2014.

Richard S Sutton and Andrew G Barto. *Reinforcement learning: An introduction*. MIT press, 2018.

Greg Brockman, Vicki Cheung, Ludwig Pettersson, Jonas Schneider, John Schulman, Jie Tang, and Wojciech Zaremba. Openai gym. *arXiv preprint arXiv:1606.01540*, 2016.

Tuomas Haarnoja, Aurick Zhou, Kristian Hartikainen, George Tucker, Sehoon Ha, Jie Tan, Vikash Kumar, Henry Zhu, Abhishek Gupta, Pieter Abbeel, et al. Soft actor-critic algorithms and applications. *arXiv preprint arXiv:1812.05905*, 2018.

Peter S Swain, Michael B Elowitz, and Eric D Siggia. Intrinsic and extrinsic contributions to stochasticity in gene expression. *Proceedings of the National Academy of Sciences*, 99(20):12795–12800, 2002.

Arren Bar-Even, Johan Paulsson, Narendra Maheshri, Miri Carmi, Erin O'Shea, Yitzhak Pilpel, and Naama Barkai. Noise in protein expression scales with natural protein abundance. *Nature genetics*, 38(6):636–643, 2006.

Wenjie Shi, Shiji Song, and Cheng Wu. Soft policy gradient method for maximum entropy deep reinforcement learning. In *Proceedings of the 28th International Joint Conference on Artificial Intelligence*, pages 3425–3431. AAAI Press, 2019.

Rui Zhao, Xudong Sun, and Volker Tresp. Maximum entropy-regularized multi-goal reinforcement learning. In *International Conference on Machine Learning*, pages 7553–7562, 2019.

Michael B Elowitz and Stanislas Leibler. A synthetic oscillatory network of transcriptional regulators. *Nature*, 403(6767):335, 2000.

Natalja Strelkowa and Mauricio Barahona. Switchable genetic oscillator operating in quasi-stable mode. *Journal of The Royal Society Interface*, 7(48):1071–1082, 2010.

Aivar Sootla, Natalja Strelkowa, Damien Ernst, Mauricio Barahona, and Guy-Bart Stan. On periodic reference tracking using batch-mode reinforcement learning with application to gene regulatory network control. In *52nd IEEE conference on decision and control*, pages 4086–4091. IEEE, 2013.

