# OpenReview forum: "Reinforcement Learning for Control with Probabilistic Stability Guarantee"
_ICLR.cc/2021/Conference — Reject_

### Official Review · AnonReviewer3 · 2020-10-27

**Rating:** 6
**Confidence:** 3

**Review:**

This paper studies the probabilistic stability guarantee of control systems. In general, hard stability guarantee is difficult with only finite samples. The authors instead focus on developing probabilistic stability conditions. High probability bound is derived in terms of the number of trajectories and the length of them. This also leads to a practical policy gradient style algorithm, which is applied to the Cartpole task with desired performance.

Overall, I think the approach makes sense. With only finite sample, the best one can hope for, without further assumptions, is to understand the probabilistic behavior. The method seems intuitive and the writing is generally clear. It is interesting to see the connection of the resulting algorithm and the classical REINFORCE algorithm. This seems to suggest that "proper" policy gradient methods can lead to stability. It would be interesting to further understand if other policy gradient style methods can be also made to exhibit such property.

While the paper contains one experiment, I think it is not enough to fully support the overall theoretical results. Since the authors already have the code available, it should be fairly easy to include more tasks with varying difficulties. In addition, since the motivation is to solve complex task, it might be interesting to understand how the overall approach scales when the dimension increases. Overall, having only one experiment makes me wonder whether the results are cherry-picked and the parameters are heavily tuned.

-----------------------------
Post rebuttal:
I thank the authors for addressing my comments and updating the manuscript. I plan to keep my positive score.

---

### Official Review · AnonReviewer4 · 2020-10-28
**A interesting step into a promising direction, but the results seem premature**

**Rating:** 6
**Confidence:** 4

**Review:**

**General overview:** The paper studies guaranteeing the closed-loop stability of a Markov decision process (MDP) using a given a policy, based on a finite number of trajectories each containing finite number of steps. Both the state and the action spaces are assumed to be subsets of finite dimensional Euclidean spaces. Particularly, the concept of mean square stability (MSS) is used which is guaranteed based on the properties of certain Lyapunov functions. Theoretical results on probabilistic MSS guarantees are provided based on finite samples. A variant of the standard REINFORCE policy gradient method is also presented which searches for a policy having MSS guarantees. Finally, numerical experiments on a simulated cart-pole problem comparing the suggested L-REINFORCE method with the soft actor-critic (SAC) off-policy RL algorithm are shown.

**Pros:**
- Though, the concept of stability is central to classical control theory, it is often neglected in the RL literature. The paper studies such a stability in continuous state and action spaces.
- Having a stochastic guarantee based on a finite sample (of finite length trajectories) is a practically motivated problem.
- The paper presents a Lyapunov function based approach and corresponding theorems to have a probabilistic guarantee on MSS.
- A policy gradient type method is introduced, as well, which searches for a policy that has the MSS property with high probability.
- Some numerical experiments are also presented comparing the results to a recent off-policy RL method.

**Cons:**
- Although, the general objective of the paper (namely, to have finite sample stochastic stability guarantees in RL) is nice and should be further studied, the paper in its current form seems premature.
- The used assumptions, for example, the ergodicity of the induced Markov chains and Assumption 2, are strong and could be hard to check in practice. For example, how should one verify Assumption 2 for a given problem? The paper should present some specific examples to show how should the conditions be checked in practice.
- The theorems, like Theorem 2, themselves are not surprising and seem like relatively simple consequences of standard concentration inequalities, such as the Hoeffding inequality.
- Also, a practically more relevant form of this result, namely to give a lower bound on M (number of trajectoris) and T (number of steps), given a confidence level, which guarantees MSS with that confidence, should also be included.
- It would be more elegant to use only one long trajectory, instead of assuming that we can generate arbitrary number of trajectories to test the stability of the system.
- The L-REINFORCE algorithm only searches for a stable controller, but it does not care about its optimality. The authors mention that an optimal controller might not be stable. Nevertheless, it would make much more sense finding the optimal stable controller, namely, the best controller (for example, with respect to the total expected reward criterion) which is also stable, in the MSS sense.
- In the light of the previos remark, it is questionable what do the experimental results demonstrate, as the SAC method of Haarnoja et al. aims at optimizing an expeted total reward criterion, while the suggested L-REINFORCE method aims at finding a stable controller (apples vs oranges).
- The paper claims that REINFORCE is a special case of L-REINFORCE. This claim is a bit strange, as the authors use a very specific choice of cost function c(s), defined on page 3 after Q2, while in the classical REINFORCE method the immediate-cost function can be arbitrary.
- Some assumptions are important, but not emphasized. For example, the Markov chain induced by the policy is ergodic with a unique stationary distribution (see page 3) should be highlighted and labeled, similarly to Assumptions 1 and 2.

---

### Official Review · AnonReviewer1 · 2020-10-28
**Finite sample stability criterion for control non-linear systems**

**Rating:** 5
**Confidence:** 3

**Review:**


### Summary

The paper describes a finite sample approximation for estimating stability of
control policy. They propose to use stability in mean squared setting with
finite sample errors on estimating Lyapunov function. The experimental results
are thin and based only on balancing a cart-pole.

### Pros

1. Take aways from the paper are intuitively reasonable. Any sampling based
   method would suffer from low samples M or high variance due to lower T in
   sequence based estimates.

2. Authors in related works highlight the importance of missing research in
   stability analysis in recent advances in RL. THe points covered in related
   work are valid and relevant

### Cons/Questions

1. I disagree that Lyapunov analysis is always impossible. Finding a Lyapunov
   candidate function is very challenging task, but it doesn't not mean there
   isn't a function. Impossible --> implies there is no such function.
2. One typically addresses the infinite horizon requirements with discounting.
   One can obtain an error bound for horizon truncation see [Section III Ref 1.]
3. From the cart pole results which a fairly linear system around the vertical
   stabilising point, the numbers for M and T look very large. Authors have not
   mentioned sampling rate for the cart-pole system, to judge what exactly the
   number T implies.

### Language

1. The paper in general would benefit from a language review. This does not
   factor in the review decision, however, authors may prefer to ease readers
   burden by getting the work reviewed for language alone.


### Ref

1. D. Ernst, M. Glavic, F. Capitanescu and L. Wehenkel, "Reinforcement Learning
   Versus Model Predictive Control: A Comparison on a Power System Problem," in
   IEEE Transactions on Systems, Man, and Cybernetics, Part B (Cybernetics),
   vol. 39, no. 2, pp. 517-529, April 2009, doi: 10.1109/TSMCB.2008.2007630.

---

### Official Review · AnonReviewer2 · 2020-10-30
**Nice work but missing references, maybe too strong assumptions?**

**Rating:** 5
**Confidence:** 4

**Review:**

The paper provides insight and results for probabilistic stability certification using a finite set of trajectories and a Lyapunov function based on a neural network plus a norm prior function. Then, a link to REINFORCE is estabilished to learn policies that are stabilising (once optimality is reached and the RL is finished).

Pros: The theoretical results seem correct. The link to REINFORCE is nice. The small example illustrates the point of stability.

Cons:

0) Some relevant citations are missing:

`Bobiti, R. V. (2017). Sampling–driven stability domains computation and predictive control of constrained
nonlinear systems. Technische Universiteit Eindhoven.`

In this thesis, a probabilistic stability result is formulated based on Lyapunov functions. Part of this work was used by some of your references (the deterministic bit) and the stochastic part was used in another missing citation:

`Marco Gallieri and Seyed Sina Mirrazavi Salehian and Nihat Engin Toklu and Alessio Quaglino and Jonathan Masci and Jan Koutník and Faustino Gomez (2019). Safe Interactive Model-Based Learning. In NeurIPS 2019 workshop on Safety and Robustness in Decision-Making`

These works are very closely related to the present paper and should be cited. It would be important to see where your results stand with respect to these related works. In particular, your sample bound based on the log probability seems almost the same as the one derived in Chapter 5 of Bobiti (2017). What is the advantage of using your bound? Is it possible to estimate the level sets as well as the Lyapunov function for local stability as done in Gallieri et al (2019)? Your function is also quite close to the one used in Gallieri et al (2019) since you have a prior norm term and you make sure it is positive definite by construction which was not done in other related work on Lyapunov networks. What are the limit of these approaches for learning and verification when the system is stochastic and how does your approach overcome them if it does?

1) The assumptions are not very clear and seem strong at times. Do you assume the origin to be an "equilibrium point" (in expectation at least). This is never stated. The definition of Lyapunov function in the introduction is wrong, you need the derivative to be negative definite in order to converge to the origin. You have that in the later definition for your special case but this is also true in the general case. Isn't it a bit too strong in Assumption 2 to ask for a single $\gamma<1$ for all values of T (all possible sequence lenghts)? Does this mean that there is effectively a contration in the system or that probability distributions converge quickly to the limit distribution? How does that fit with the claim that the system under analysis is "highly stochastic" (made in the introduction)?

2) It is claimed that the system to deal with is "highly stochastic". Then the approach is tested on a cartpole, which is deterministic.
Could you provide an example of a system that is "highly stochastic"?  I think it would then make sense to try your approach also on such a system and not just a cartpole.

3) Does your result hold for any distribution of the initial states? You have a bound on the probability mass but it seems like the distribution does not affect your results in any ways?

4) Can you scale the approach to a more complex example then the cartpole? It seems rather simple and deterministic. Your technique seems like an overkill for the cartpole. That can be solved by a linear deterministic controller.
 Since you compare to SAC and use REINFORCE, you should have all the machinery in place to address an example that cannot be controlled via LQR, right? There shouldn't be any further coding required...

Notation is a bit hard to follow at times, in particular when discussing closed-loop and open loop distributions the same symbols are used. There are also a few typos that should be checked: Lemma 1 "denote", above Lemma 3 "affect".

---

### Decision · Program_Chairs · 2021-01-07
**Final Decision**

**Decision:**

Reject

**Comment:**

Although the reviewers like the general idea of the paper, there are concerns regarding the clarity of the statements, especially in stating the main assumptions, referring to related work, and how well the experiments support the results of the paper. Although the authors' long response addressed some of the issues/comments raised by the reviewers, not all of them are convinced that the paper carries enough novelty and is ready for publication. I would suggest that the authors take the reviewers' comments into account, revise the paper, and make it ready to be submitted to an upcoming conference.